# Coda: An End-to-End Neural Program Decompiler

**Cheng Fu, Huili Chen, Haolan Liu**
UC San Diego
{cfu,huc044,hal022}@ucsd.edu

**Xinyun Chen**
UC Berkeley
xinyun.chen@berkeley.edu

**Yuandong Tian**
Facebook
yuandong@fb.com

**Farinaz Koushanfar, Jishen Zhao**
UC San Diego
{farinaz,jzhao}@ucsd.edu

## Abstract

Reverse engineering of binary executables is a critical problem in the computer security domain. On the one hand, malicious parties may recover interpretable source codes from the software products to gain commercial advantages. On the other hand, binary decompilation can be leveraged for code vulnerability analysis and malware detection. However, efficient binary decompilation is challenging. Conventional decompilers have the following major limitations: (i) they are only applicable to specific source-target language pair, hence incurs undesired development cost for new language tasks; (ii) their output high-level code cannot effectively preserve the correct functionality of the input binary; (iii) their output program does not capture the semantics of the input and the reversed program is hard to interpret. To address the above problems, we propose Coda[1], the first end-to-end neural-based framework for code decompilation. Coda decomposes the decompilation task into of two key phases: First, Coda employs an instruction type-aware encoder and a tree decoder for generating an abstract syntax tree (AST) with attention feeding during the code sketch generation stage. Second, Coda then updates the code sketch using an iterative *error correction machine* guided by an *ensembled neural error predictor*. By finding a good approximate candidate and then fixing it towards perfect, Coda achieves superior performance compared to baseline approaches. We assess Coda's performance with extensive experiments on various benchmarks. Evaluation results show that Coda achieves an average of 82% program recovery accuracy on unseen binary samples, where the state-of-the-art decompilers yield 0% accuracy. Furthermore, Coda outperforms the sequence-to-sequence model with attention by a margin of 70% program accuracy. Our work reveals the vulnerability of binary executables and imposes a new threat to the protection of Intellectual Property (IP) for software development.

## 1 Introduction

Decompilation is the process of translating a binary executable to the corresponding high-level code. This technique has been widely used in various security applications, such as malware analysis and vulnerable software patching [1, 2]. Malicious attackers can also use decompilers to reverse engineer (RE) the commercial off-the-shelf (COTS) software products and reproduce it for illegal usage [3]. Decompilation is a challenging task since the semantics in the high-level programming language (PL) is obliterated during compilation. Existing decompilers are language-specific and incur tremendous engineering overhead when extending to new PLs. Furthermore, they fail to preserve the semantic information in the target high-level PL (see Appendix F), thus the output is hard to interpret.

It is intuitive that decompilation can be formulated as a general program translation task. Recently, an increasing number of neural network (NN)-based approaches have been proposed to tackle natural language translation problems. For instance, sequence-to-sequence (Seq2Seq) based models achieve

the state-of-the-art performance on program translation [4, 5]. We identify three main subroutines in code decompilation: (i) learning control dependency from the connections between basic blocks in the low-level code; (ii) learning data dependency from the register usage and memory access; (iii) learning the grammar of the target PL. A straightforward neural-based solution is to use an autoencoder-decoder for translating the low-level program to the high-level code. Katz et al. [6] present a Recurrent Neural Network (RNN)-based method for decompilation. However, we observe that the naive Seq2Seq models are not suitable for decompilation due to the following reasons. First, the inputs to the decompiler are structured low-level statements[2] that have different construction formats (e.g., number and type of operands). Processing the program as sequence inputs ignores the statement boundaries, thus breaks the modular property of the input program. Second, the output program of the Seq2Seq model has a lower probability of capturing the grammar of the target PL since the output is sequentially generated without explicit boundaries. Third, the three subroutines mentioned above are entangled together in the Seq2Seq model, making the learning process hard.

In this work, we propose Coda, a neural program decompilation framework that resolves the above limitations. The requirement to yield a perfect program recovery is very hard to fulfill using a single autoencoder, especially for long programs. As such, Coda decomposes decompilation into two sequential phases: *code sketch generation* and iterative *error correction*. By finding a good approximate program and then iteratively updating it towards the perfect solution using dynamic information, Coda engenders superior performance compared to the single-phase decompilers.

■ **Phase 1.** Coda uses an instruction type-aware encoder and a abstract syntax tree (AST) decoder for translating the input binary into the target PL. Our encoder deploys separate RNNs for different types of statements, thus the statement boundaries are preserved. Furthermore, the control and data dependency in the input program are translated to the connections between the hidden states of corresponding RNNs. The output from the AST decoder maintains the dependency constraints and statement boundaries using terminal nodes, which facilitates learning the grammar of the target PL.
■ **Phase 2.** In this stage, Coda employs an RNN-based error predictor (EP) to identify potential

prediction mistakes in the output program from Phase 1. Ensembling method can be used to boost the performance of the error prediction. The EP is used to guide the iterative correction of the output program. Unlike traditional decompilers which utilize only the syntax information from the input, Coda leverages the Levenshtein edit distance (LD) [7] between the compilation of the updated output program and the ground-truth low-level code to prevent false alarms induced by the EP.

Empowered by the two-phase design, Coda achieves an average program accuracy of 82% on various benchmarks. While the Seq2Seq model with attention and the commercial decompilers yield 12% and 0% accuracy, respectively. We demonstrate that Coda's output preserves both the functionalities and the semantics. In summary, this paper makes the following contributions:

• Presenting the first neural-based decompilation framework that maintains both semantics and functionalities of the target high-level code.
• Incorporating various design principles to facilitate the decompilation task. More specifically, Coda deploys instruction type-aware encoder, AST tree decoder, attention feeding, iterative error correction that leverages both the static syntax and dynamic information.
• Enabling an efficient *end-to-end* decompiler design. Coda can be easily generalized to RE executable in a different hardware instruction set architecture (ISA) or PL with negligible engineering overhead.
• Corroborating Coda's general applicability and superior performance on various synthetic benchmarks and real-world applications.

This is the first paper that provides a holistic and effective solution to the decompilation problem using deep learning. Our work sheds new light on the vulnerability of open sourcing binary executables without any protection. More specifically, we show that the attacker can recover interpretable high-level code with correct functionality from the binary file, which imposes a significant threat on the Intellectual Property (IP) of the program developer.

## 2   Program Decompilation Problem

We introduce the background of low-level code construction and potential challenges in decompilation in Section 2.1. The formal definition of code decompilation and our threat model is given in Section 2.2 and Section 2.3, respectively.

## 2.1 Preliminaries and challenges

Contemporary software development consists of the following steps: high-level programming, code compilation, deploying the obtained binary files to the pertinent hardware. During the execution, a sequence of instructions is carried out on the hardware. There are three main instruction types, namely, *memory*, *arithmetic*, and *branch* operations. Different instruction types feature different *instruction fields*, indicating various types and numbers of operands. Figure 1 (a) shows an example of the high-level code snippet and the corresponding low-level code. `Line 0` is a memory instruction which fetched a word into register `$1` from the memory address computed from register `$fp` and `24`. `Line 3` is an arithmetic operation which multiplies the value stored in `$1` and `$2`. `Line 8` refers to an unconditional branch requiring one operand as opposed to three. Note that `lw,mul,j` are the *opcodes* of the instructions. Program decompilation is challenging since there are two types of dependencies existing in the low-level program that shall be preserved by the decompiler.

■ **Intra-statement dependency.** Each instruction has a strict structure restriction on the operands as required by the grammar of the low-level ISA. For example, in the instruction `lw $2,8($fp)`, the first and the third operand represent registers while the second operand is an instant value.

■ **Inter-statement dependency.** Besides the constraints in a single instruction, *control flow* and *data dependency* exist across multiple instructions. For instance, line 2 and line 3 has data dependency since the `mul` operation needs to consume the value from the load destination register.

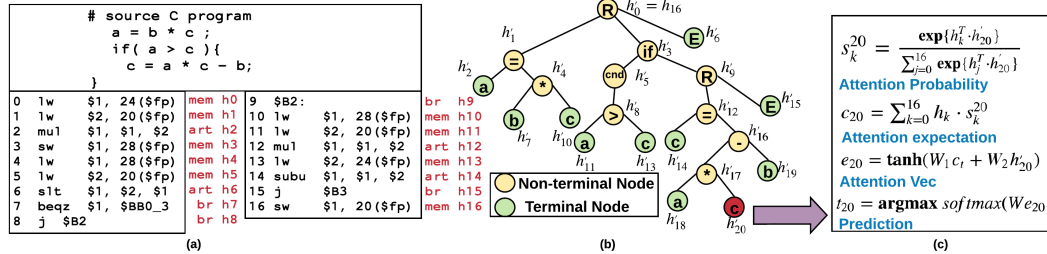

**Figure 1:** (a) Example low-level assembly code snippet and its corresponding high-level C program. The red line indicates the instruction type and its encoded hidden state. (b) The expanding nodes from the AST decoder. (c) The red node is an example of how the prediction is computed.

## 2.2 Problem definition

We define the task of *Program Decompilation* as follows:

**Problem Decompilation Definition:** *Let $P$ denote an arbitrary program in the high-level language and $\Gamma$ denote the compiler. Given the low-level code $\phi = \Gamma(P)$, the mission of decompilation is to develop a decompiler $\Gamma^{-1}$ that satisfies $\Gamma(P) = \Gamma(P')$ where $P' = \Gamma^{-1}(\phi)$.*

We observe that traditional decompilers such as RetDec or Hex-Rays are only targeted to maintain the functionality of the binary code during decompilation. Coda is motivated to address the above limitation by recovering a high-level program with both correct functionality and semantics. Besides, we identify two types of constraints of the high-level program that can be explored to verify the correctness of the decompiler's output.

**Input-output Behavior Constraint:** *Given a set of input-output pairs $\{(I^k, O^k)\}_{k=1}^{K}$ where $O^k = \phi(I^k)$ is obtained by executing the low-level program $\phi$, the decompiler shall output a program $P'$ such that $\phi'(I^k) = O^k$ for every $k \in 1, ..., K$ where $\phi' = \Gamma(P')$.*

**Compilation Matching Constraint:** *The ideal LD between the compilation result $\phi'$ of the correctly recovered program and the input low-level code $\phi$ is zero under the same compiler configuration.*

## 2.3 Threat Model

We assume the attacker has the following information: (i) the *compiler configuration* that is used to generate the input program; (ii) the interface of *static/dynamic libraries* included in the high-level code; (iii) the disassembler for the pertinent hardware. The above information can be easily obtained using binary analysis techniques in previous work [8–10]. Our objective is to RE a high-level program that depicts the correct computation graph (control and data dependency) and preserves semantic information and functionality as the source high-level program. Reconstruction of data types [3], finding the function entry point in binary [10, 11, 8] or reconstruct meaningful variable names [12] are different research directions that have been studied in prior works.

# 3 Coda Overview

Figure 2 shows the global flow of Coda. Coda framework consists of two key phases: (i) High-level code sketch generation and (ii) iterative error correction. We detail these two phases as follows.

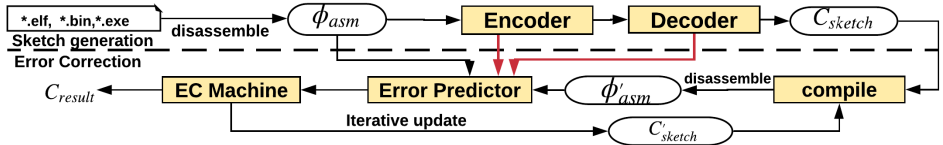

**Figure 2:** The global flow of Coda decompilation. The denoising and tokenization steps are omitted in this figure for simplicity (See Appendix A.1).

## 3.1 Code Sketch Generation

We employ the neural encoder-decoder architecture for generating the sketch code from the low-level program $\phi$. In this paper, the encoder takes the assembly program generated from the disassembler as the input and output an AST that can be equivalently converted to the high-level program. We discuss the key modules of Coda's code sketch generation below.

■ **Instruction-type Aware Program Encoding.** Coda employs the N-ary Tree-LSTM presented in [13] as the input encoder to handle different instruction types, namely, memory, arithmetic, and branch. More specifically, each statement in the input low-level program is fed to the designated LSTM that handles the corresponding instruction type for encoding.

■ **Tree Decoder for AST Generation.** We observe that PLs have more rigorous restrictions on the syntax and semantics compared to natural languages. Coda opts to use tree decoder for AST generation because of the following advantages: (i) The code statement boundary is naturally preserved by the tree decoder using the terminal node representation. (ii) The nodes that are connected in the AST indicate that their corresponding statements in the input program have dependency constraint. Note that the spatial distance for these statements in the program might be large. (iii) The error propagation problem during code generation is mitigated using the tree decoder compared to sequential generation. (iv) AST representation facilitates the verification of syntax restriction.

■ **Attention Feeding.** Our evaluation result shows that code sketch generation is ineffective without attention mechanism (achieving a token accuracy of only 55%). Coda applies parent attention and input instruction attention feeding mechanism [14, 15] that feed the information of the parent node and the input nodes during node expansion performed by the decoder.

## 3.2 Iterative Error Correction

The output AST from the code sketch generation phase might contain prediction errors. As such, we construct an error predictor (EP) and an iterative *error correction machine (EC machine)* as shown in Figure 2. Specifically, we freeze the autoencoder-decoder from the previous stage and reuse them to generate the states of the input nodes $h_k, k = 0, ..., K$ and output nodes $h_t, t = 0, ..., T$. Here, $K$ and $T$ denote the total number of input states and output states from sketch generation stage. These states ($h_k$ and $h_t$) are fed as the input to the EP. Furthermore, Coda leverages compiler verification to remove false alarm made by the EP. Note that the input-output behavior of the ground-truth binary executable can also be used as constraints that eliminate false alarms from EPs. To push the performance even further, we ensembled multiple EPs to cover more errors in the decompiled code.

■ **Iterative Error Correction machine.** The output of the ensembled EP (containing the location and error type information in the code sketch) is passed to the EC machine. Note that the EC machine prioritizes the potential correction strategies based on the confidence scores obtained from the EP. During each iteration of the error correction process, Coda first corrects a single error and validate the resulting high-level code sketch by checking the LD between the compiled code sketch and ground-truth as mentioned above. If the error correction is successful, Coda proceeds to the next iteration where EP generates new guidance for the EC machine.

# 4 Methodology

In this section, we detail two key phases of Coda's design as shown in Figure 2: autoencoder based code sketch generation, and neural-based iterative error correction.

## 4.1 Autoencoder-based Code Sketch Generation

We introduce the main modules of Coda's code sketch generation phase as follows.

■ **Instruction-aware encoding.** The computation flow of Coda's input program encoding is shown in Equation (1) where the subscript $n$ refers to $n_{th}$ instruction statements. To capture the intuition of learning the intra and inter-dependency of the instruction statements as discussed in Sec 2, Coda employs an N-ary Tree Encoder [13] which is suitable for encoding task where the children are structured. The input states are fed into the N-ary Tree Encoder with a consistent order of the corresponding instruction type. As such, the intra-statement dependency can be effectively captured. Particularly, Coda designates a specific N-ary encoder for each instruction type, i.e., memory, branch and arithmetic instructions ($LSTM_i$ where $i \in \{mem, br, art\}$). Note that in Coda's code encoding process, each non-terminal node has at most 4 children, consisting of the embedded states of up to 3 operands in the current instruction ($h_i^{op}, c_i^{op}$ where $i = 0, 1, 2$), and the context vector of the previous instruction ($h_n, c_n$). Coda encodes the input instructions with the maximal number of operands (i.e., 3) and pads short statements with zero states. The input $x$ is the embedding of the instruction opcode of the current statement. The basic block header (e.g., $\$B2$ in line 9 of Figure 1 (a)) are also handled as branch instructions.

$$(h_{n+1}, c_{n+1}) = LSTM_i(([h_n; h_0^{op}; h_1^{op}; h_2^{op}], [c_n; c_0^{op}; c_1^{op}; c_2^{op}]), \ x), i \in \{mem, br, art\} \quad (1)$$

■ **Binary Abstract Syntax Tree decoder.** The output states of the last instruction in the low-level code is used as the input to Coda's tree decoder for AST generation. Non-leaf nodes in general AST structures may have multiple children, which complicates the high-level code generation process since the number of children varies for different nodes. To address this uncertainty in the decoding stage, Coda generates a binary tree in Left-Child Right-Sibling representation which is equivalent to the target AST output. As a result, each sub-tree in the AST output has a regulated structure that is consistent with the root. We deploy two LSTMs that predict the left and the right child of the current node separately. The states $(h, c)$ from a given AST node will be fed into the left/right LSTM to generate the left/right child, as shown in Equation (2) and (3). These two expanded nodes will become the new parent nodes to generate its children using the left/right LSTM. The obtained binary AST tree will be transformed back to its equivalent AST tree. Note that Coda's output AST does not contain the statement ending token as the termination is naturally represented by the terminal nodes. For example, a complete statement $a = b * c$ can be recovered from the AST subtree without an explicit ending token as shown in Figure 1. The state transition equations of Coda's AST decoder are shown as follows:

$$(h_L, c_L) = LSTM_L((h, c), [Ho_t; e_t]) \quad (2)$$

$$(h_R, c_R) = LSTM_R((h, c), [Ho_t; e_t]) \quad (3)$$

Here, the subscript $t$ denotes the current expanding node $N_t$ in the output AST. The symbols $o_t$ and $e_t$ indicate the predicted token value and the attention vector (explained later) of the node $N_t$. $H$ is a trainable token embedding matrix with dimension $d \times V$, where $d$ and $V$ are the embedding dimension and the vocabulary size of the high-level PL, respectively.

■ **Input Instruction and Parent Attention Feeding.** To make better use of the information encoded from the input program and the parent context of the current expanding node, we employ instruction and parent attention feeding during AST decoding [16, 14]. Intuitively, predicting the current node while leveraging the relevant information from the input instructions and the node's parent provides a richer context for high-level code generation. Parent attention feeding is performed using Equation (2) and (3) during the state transition of the AST decoder. As for input instruction attention feeding, we first compute the probability that a node $N_k$ in the input program corresponds to the expanding node $N_t$ as shown in Equation (4). Coda's input instruction attention is obtained from the expectation value of the hidden states of all nodes in the input program as shown in Equation (5).

$$s_k^t = P(N_k|N_t) \propto \mathbf{exp}\{h_k^T \cdot h_t\} \quad (4)$$

$$c_t = \mathbb{E}[h_k|N_t] = \sum_{k=0}^{K} h_k \cdot s_k^t \quad (5)$$

$c_t$ is then incorporated into the hidden state of the current node $h_t$ using Equation (6) where $W_1$ and $W_2$ are two trainable matrices with dimension $d \times d$, resulting in the attention vector $e_t$ of the current node. The final prediction output $o_t$ of the current expanding node is then be computed from the linear mapping of $e_t$ as shown in Equation (7). $W_{out}$ is a trainable matrix of size $V \times d$.

$$e_t = \mathbf{tanh}(W_1 c_t + W_2 h_t) \quad (6)$$

$$o_t = \mathbf{argmax} \ softmax(W_{out} e_t) \quad (7)$$

## 4.2 Neural-based Iterative Error Correction

We propose iterative Error Correction as the second phase of Coda framework to further improve the quality of decompilation as discussed in Sec. 3.2. There are two key modules in this stage: an ensembled neural EP and an Iterative *EC machine*. We characterize possible errors in Coda's code sketch into three types: (i) Nodes in the AST may be mispredicted to other tokens. For example, the 'while' might be misclassified into 'if' token. (ii) A redundant line of code. (iii) A Missing line of code. For error (i), the EP shall output the correct token value to guide the EC machine for updating the node. For error (ii) and (iii), the EC machine removes/randomly adds a non-terminal node with leaf children in the predicted error location, thus converting the error type into a misprediction error (i). (See Appendix B for details) Equation (8) shows the hidden state transition of Coda's EP. We deploy the fixed autoencoder from phase 1 followed by gated recurrent units (GRUs) with attention as the EP's architecture. Given the ground-truth input ($\phi$) and the compiled code sketch ($\phi'$), the EP returns the error status ('0/1') and the error types for each node in the output AST. The input to the GRU consists of two parts: (i) the hidden state of the parent node ($h_{t-1}^{EP}$); and (ii) the concatenation of the hidden states (denoted by $h_t^{\phi}$ and $h_t^{\phi'}$) obtained by forwarding $\phi$ and $\phi'$ to the autoencoder.

$$h_t^{EP} = GRU(h_{t-1}^{EP}, [h_t^{\phi}; h_t^{\phi'}]) \tag{8}$$

The attention layer in EP following the mechanism discussed in Sec. 4.1. Particularly, the input to the attention layer $h_t$ in Equation( 4) is now replaced by the hidden state $h_{EP}$ of the current node. The state of source input ($h_k$) is substituted with the combination of the encoded states $h_k^{\phi}$ and $h_k^{\phi'}$. Furthermore, Coda ensembles multiple EPs to cover larger error space. The correction suggestion provisioned by the EP is accepted if and only if the LD between the golden low-level code and the compilation of the updated code sketch does not increase.

The workflow of Coda's iterative *EC machine* is shown in Algorithm 1. The detail of the function $FSM\_Error\_Correct$ in line 9 is presented in Appendix B.

---

**Algorithm 1** Workflow of iterative EC Machine.

---

**INPUT:** $N_{EP}$ **Ensembled Error Predictors** $EP$; **Source assembly** $\phi$; **Decompiled Sketch program** $P'$; **Compiler** $\Gamma$; **Maximum iterations** $S_{max}$ **and steps in each iteration** $c_{max}$;

**OUTPUT: Error corrected program** $P_f'$.

1:   $s_i \leftarrow 0$
2: **while** $s_i < S_{max}$ **do**
3:      $Q \leftarrow [\,], \phi' = \Gamma(P'), \Delta' \leftarrow Edit\_loss(\phi, \Gamma(P'))$
4:      **if** $\Delta' = 0$ **then break**
5:      $Q \leftarrow EP_i(P')$ for $i = 1,...,N_{EP}$      // Attach all the detected error to queue $Q$
6:      $\widetilde{Q} \leftarrow Prob\_sort(Q, c_{max})$      // Rank $Q$ using output probabilities, keep $c_{max}$ results.
7:      **while** $\widetilde{Q}$ is not empty **do**
8:          $err, node \leftarrow \widetilde{Q}.pop()$
9:          $P_t' \leftarrow FSM\_Error\_Correct(P', err, node)$      // correct the error in the program
10:         $\Delta = \Delta' - Edit\_loss(\phi, \Gamma(P_t'))$
11:         **if** $\Delta \geq 0$ **then**
12:             $P' \leftarrow P_t'$
13: **Return:** $P_f' \leftarrow P'$

---

## 5 Evaluation

### 5.1 Experimental Setup

We assess the performance of Coda on various synthetic benchmarks with different difficulty levels and real-world applications as summarized in Table 1 (Stage 1) and Table 2 (Stage 2). Given the binary executable as the input, we use an open-source disassembler [17, 18] for MIPS [19] and x86-64 [20] architecture to generate the corresponding assembly code that is fed to Coda.

∎**Benchmarks.** We describe the four main tasks in our evaluation as follows.

**(i) Karel.** Karel [21] is a C-based library that can be used to control the movement of a robot in a 2D grid and modify the status of the environment. The assembly description of Karel programs has only callback functions (no arguments) and global control flags as shown in Appendix E. As such, Karel is suitable to evaluate Coda's capability of recovering the control flow graphs (CFGs) of the source code (see possible CFGs in Appendix A.2).

**(ii) Math Library (Math).** We generate synthetic benchmarks using `math.h` library [22] to assess Coda's performance for recovering both data and control dependencies.

**(iii) Normal Expression (NE).** Common operations such as $"+, -, *, \backslash, \|, \gg, \&, ==, \wedge"$ are the main components of NEs in high-level PL. We observe that reconstructing normal expressions is more difficult compared to function calls since the former one has less explicit structures.

**(iv) Composition of Functional Calls and Normal Expressions (Math+NE).** We also construct synthetic benchmarks consisting of both NEs and library functions calls. The dataset is constructed by replacing the variables in NE with the return value of a random math function (see Appendix E).

**(v) Real-world implementations.** We test Coda's performance on real-world projects: (1) neural network construction programs in pytorch C++ API [23] (2) Hacker's Delight loop-free programs [24] provided in [25]. These programs are used for encoding complex algorithms as small loop-free sequences of bit manipulating instructions.

■ **Training Data Generation.** To build the training dataset for stage 1, we randomly generate 50,000 pairs of high-level programs with the corresponding assembly code for each task. The program is compiled using `clang` with configuration `-O0` which disables all optimizations. The subscript $S$ and $L$ in Table 1 denotes short and long programs with an average length of 15 and 30, respectively. The tree representation of each statement in the high-level code has a maximum depth of 3.

The training dataset for the error correction stage is constructed by injecting various types of errors into the high-level code. In our experiments, we inject $10 \sim 20\%$ token errors whose locations are sampled from a uniform random distribution. To address the class imbalance problem during EP training, we mask $35\%$ of the tokens with error status '0' (i.e., no error occurs) when computing the loss. Detailed statistics and examples of the dataset can be found in Appendix A.2 and E.

■ **Error Predictor Training.** We manually inject three types of errors (Sec. 4.2) into the ground-truth code sketch and set the training target for each EP to the corresponding error types. Note that the EP also learns to output the correct substitution token when the target is the misprediction error type.

■ **Metrics.** We evaluate the performance of the Coda using two main metrics: *token accuracy* and *program accuracy*. Token accuracy is defined as the percentage of the predicted tokens in high-level PL that match with the ground-truth ones. Program accuracy is defined as the ratio between the number of predicted programs with 100% token accuracy and the number of total recovered programs.

## 5.2 Results

**Performance of Sketch Generation.** Coda yields the highest token accuracy across all benchmarks (96.8% on average) compared to all the other methods as shown in Table 1. The NE task appears to be the hardest one while Coda still engenders 10.1% and 80.9% margin over a naive *Seq2Seq* model with and without attention, respectively. More importantly, the result demonstrates that our *Inst2AST+Attn* method is more tolerant of the growth of the program length compared to the *Seq2Seq+Attn* baseline. We hypothesize that this is because *Inst2AST* with attention focuses on the states of each instruction as a whole instead of every input token. As such, it is less sensitive to the growth of assembly token length. Note that Coda achieves higher token accuracy on Math+NE benchmarks compared to NE ones. This is due to the fact that the assembly description of function calls has a prologue of argument preparation [11] that is easy to identify than NE which directly operates on variables.

We observe that the majority of the token errors are misprediction in the sketch, especially when the program size is large. Besides, the sketch may have missing or repetition statements. The imperfection of code sketch generation motivates the design of the error correction stage in Coda. Our empirical results show that in the recovered code sketch from Stage 1 has very few syntax errors that lead to decompilation failure. This further validates the capability of Coda to automatically learn the syntax structure of the high-level language. A small portion of syntax errors exists in the sequence decoding baseline and we use a script to check and fix these syntax bugs. The token accuracy reported in Table 1 is measured before the script checking. Token errors that do not result in compilation failure are corrected in the Phase 2.

**Performance of Error Correction.** Table 2 summarizes the performance of Coda's iterative error correction. We feed the recovered code sketch with imperfect token accuracy generated from stage 1 to the pretrained EP. Recall that the EP consists of the fixed autoencoder from stage 1 and a GRU layer. Here, EPs that reuse the *Seq2Seq+attn* and *Inst2AST+attn* sketch generator are denoted as $\text{EP}_{s2s}$ and $\text{EP}_{i2a}$, respectively. We set $S_{max} = 30$ and $c_{max} = 10$ for *EC machine* in Algorithm 1.

**Table 1:** Token accuracy (%) comparison between Coda and alternative methods for code generation. Columns 1-2 denotes the *Seq2Seq* baseline. The last two columns denote the instruction-aware encoding (Inst) and AST decoding (AST) methods of Coda with and without attention (Attn) mechanism. The combination of a sequence-based model with Inst or AST is shown in Columns 3-4.

| Benchmarks | Seq2Seq | Seq2Seq+Attn | Seq2AST+Attn | Inst2seq+Attn | Inst2AST | Inst2AST+Attn |
|---|---|---|---|---|---|---|
| $Karel_S$ | 51.61 | 97.13 | 99.81 | 98.83 | 74.80 | **99.89** |
| $Math_S$ | 23.12 | 94.85 | 99.12 | 96.20 | 56.29 | **99.72** |
| $NE_S$ | 18.72 | 87.36 | 90.45 | 88.48 | 55.59 | **94.66** |
| $(Math+NE)_S$ | 14.14 | 87.86 | 91.98 | 89.67 | 56.62 | **97.90** |
| $Karel_L$ | 33.54 | 94.42 | 98.02 | 98.12 | 64.42 | **98.56** |
| $Math_L$ | 11.32 | 91.94 | 96.63 | 93.16 | 45.63 | **98.63** |
| $NE_L$ | 11.02 | 81.80 | 85.92 | 85.97 | 46.43 | **91.92** |
| $(Math+NE)_L$ | 6.09 | 81.56 | 85.32 | 86.16 | 43.77 | **93.20** |

A single EP achieves 66% ($EP_{s2s}$) and 69% ($EP_{i2a}$) accuracy on average across benchmarks for predicting the error type in the sketch programs. Note that we only consider the prediction of the first error due to the iterative nature of the EC machine.

When ensembling 10 EPs ($N_{EP} = 10$), the detection rate of first error can be enhanced to 84% and 89% on average for $EP_{s2s}$ and $EP_{i2a}$, respectively. Note that $EP_{i2a}$ achieves a higher accuracy on error prediction across benchmarks compared to $EP_{s2s}$. That is because the component of the $EP_{i2a}$, i.e., *Inst2AST+attn*, achieves a better token accuracy compared to *Seq2Seq+attn* in $EP_{s2s}$.

The ensembled EPs will guide our iterative EC machine as detailed in Algorithm 1. Coda's EC machine increases the program accuracy from 12% to 61% and from 30% to 82% on average for *Seq2Seq+Attn*-based and *Inst2AST+attn*-based code sketch generation, respectively. In summary, Coda's best configuration (*Inst2AST+attn* with EC) achieves an average of 82% final program accuracy while the *Seq2Seq* model with or without attention approach yields 12% and 0%, respectively.

**Table 2:** (i) First error prediction accuracy with various ensembled number of ensembled EPs. (ii) Program accuracy before and after error correction (EC) when $N_{EP}$=10. Note that $N_{EP}$ stands for the number of ensembled EPs and model refers to the architecture of sketch generation.

| BenchMarks | (i) First Error Detection Rate (EP,$N_{EP}$) | | | | | | (ii) Befor EC | | After EC | |
|---|---|---|---|---|---|---|---|---|---|---|
| | $s2s,1$ | $i2a,1$ | $s2s,5$ | $i2a,5$ | $s2s,10$ | $i2a,10$ | $s2s$ | $i2a$ | $s2s$ | $i2a$ |
| $Math_S$ | 69.6 | 74.1 | 84.9 | 88.5 | 91.4 | 94.2 | 40.1 | 64.8 | 91.2 | **100.0** |
| $NE_S$ | 64.2 | 67.6 | 76.0 | 79.2 | 83.5 | 88.7 | 6.6 | 12.2 | 53.0 | **78.6** |
| $(Math+NE)_S$ | 65.1 | 67.3 | 78.4 | 84.4 | 83.6 | 90.1 | 3.5 | 43.2 | 63.6 | **89.2** |
| $Math_L$ | 65.4 | 71.7 | 80.9 | 83.1 | 87.5 | 91.3 | 21.7 | 51.8 | 83.9 | **99.5** |
| $NE_L$ | 60.3 | 64.7 | 71.6 | 76.5 | 78.1 | 84.5 | 0.2 | 2.6 | 33.1 | **56.4** |
| $(Math+NE)_L$ | 61.0 | 66.5 | 73.9 | 77.5 | 80.2 | 85.3 | 0.1 | 4.9 | 38.3 | **67.2** |

**Results on Real-world Applications.** We assess Coda on two real-world applications: Pytorch C++ API-based [23] model architecture construction and bit twiddling hack in Hacker's Delight [24]. The model definition and the bit twiddling task programs consist of a sequence of function calls and a sequence of loop-free normal expressions, respectively. Examples of these two applications are given in Appendix E. Coda achieves 100% program accuracy across all benchmarks for these two tasks.

**Comparison to Previous Works.** We demonstrate that Coda outperforms two state-of-the-art decompilers: RetDec [26] open-source tool and sequence-to-sequence based decompiler [6]. The output from RetDec has a large LD to the ground-truth low-level code after compilation. Furthermore, the high-level program recovered by RetDec fails to preserve the functionality of the input and is hard to interpret. (See Appendix F for examples). The *Seq2Seq*-based approach proposed in [6] takes a sequence of bytes or bits directly from the binary executable as the input. We re-implement their technique and assess its performance on Math+NE synthetic benchmarks. Empirical results show that their *Seq2Seq*-based method achieves 11% token accuracy and 0% program accuracy on average.

## 5.3 Discussion

We discuss the factors that might influence the performance of Coda framework as follows.

■ **Recover complex ISA.** We identify that the performance of Coda decreases when the input low-level code is buit from more complicated ISA, such as x86-64 (details shown in Appendix D). This is mainly because: (i) x86-64 has more advanced instruction types that support different levels of

granularity for memory read/write while MIPS supports only 32-bits read/write operations. As such, the number of input token types in x86-64 is much larger than the one in MIPS ISA when compiling the same high-level program; (ii) the branch flag is not visible as part of the branch instructions in x86-64 since it is stored in the condition register. In MIPS ISA, Coda can extracts the branch flag directly from the input instruction. Therefore, it is harder to recover the control dependencies in x86-64 compared to MIPS.

■ **Recover complicated structures.** In Section 5.2, we evaluate Coda on synthetic benchmarks that have the same program components as the previous decompiler works [27]. These benchmarks include function calls, normal expressions, nested control graphs, variables with different types and data dependencies. Complex data structures/control graph/static library reliance impose greater challenges to the program decompilation task. Data type and structure identification is an individual research direction which has been widely studied in previous works [3, 28]. Coda is sufficient to resolve real-world applications such as Pytorch API or Hacker's delight applications. Combining Coda with these works can recover more complicate programs.

■ **Recover long programs.** We assess Coda on programs (NE and Math+NE) with the average code length ($L$) of 45 and 60. The token accuracy in Phase 1 drops by -5.4%/-13.5% (Inst2AST) on average compared to the results on benchmarks with $L = 30$. We identify two challenges to decompile long programs: (i) Unlike natural language with period as the end of sentence, there is no clear boundary to divide assembly code. The length of the input tokenized assembly grows to a very large value (Appendix A.2). (ii) Also, training the auto-encoder in Coda is more challenging for long encoding sequences due to the limited GPU memory.

# 6    Related Work

**Conventional Program Decompilation.** There has been a long line of research on reverse engineering of binary code [29–31, 8, 10, 27, 2, 32]. Many decompilers such as Phoenix [27], Hex-rays [33] and RetDec [26, 34] (the most recent one) have been developed. Other works, such as TIE [3] or RE-WARDS [35], target at reconstructing the correct variable types, which is different from the objective of Coda. Learning-based methods have been proposed for identifying function entry point [8, 10, 11] for disassembling binary code. These methods are orthogonal to Coda and can be integrated into our framework to tackle a wider range of decompilation tasks. To the best of our knowledge, no practical deep learning-based techniques have been proposed for program decompilation.

**Neural Networks for Code Generation.** Neural networks have been used for code generation in prior works [36–38, 14, 39]. Instead of recovering the high-level program from the corresponding assembly code, these works synthesize the program from the collected input-output pairs [40, 39], natural language [37], or other domain-specific languages [14, 4]. In [6], they use a sequence-to-sequence model for decompilation with direct Byte-by-byte sequence input which yields a low accuracy as shown in Sec. 5. Coda demonstrates the first effective program decompilation framework.

**Neural Networks for Error Correction.** The idea of iteratively fixing errors in the program using neural networks has been proposed [41–43]. In [42], they suggest using GRUs for embedding the execution trace in order to identify bugs in the target program. DeepFix [41] deploys an autoencoder to fix typos that leads to a compilation failure. Note that the error correction stage of Coda has a different objective from the above works. More specifically, we use an autoencoder-based error predictor to identify the token errors in output from the code sketch generation stage.

# 7    Conclusions and Future Work

In this paper, we present Coda, the first neural-based decompilation framework that is corroborated to preserve both the semantics and the functionality of the high-level program. Coda consists of two key phases for program RE. First, Coda generates the high-level program with a high token accuracy leveraging an instruction-aware encoder and an AST decoder network architecture with attention. Next, Coda iterative correct errors with the guidance of the ensembled EP which further improves Coda's token and program accuracy. Extensive experiments on various benchmarks corroborate that Coda outperforms the Seq2Seq model and traditional decompilers by a large margin. We believe that our work is a milestone for program security and decompilation.

Meanwhile, we observe that several challenges remain in our current framework that can be addressed in the future work: (i) There are no explicit ending symbols in decompilation task. Future research can tackle this issue to RE large-size binary file. (ii) Previous works on the identification of more complicated data structures can be incorporated into Coda to RE more complicated applications.

## Footnotes

[1] *Coda* is the abbreviation for *CodeAttack*.

[2]   We refer each line of the code as a statement.

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
