[Supplementary Material]

# A    Experiment Setup and Benchmark Details

We ran our experiments on Amazon EC2 using `p3.4xlarge` instance which contains Nvidia Tesla V100 GPUs with 16GB main memory.

## A.1    Denoising Process

We start the tokenization of the low-level assembly from the beginning of the program function to be reversed, e.g., 'func:' in assembly of our case. All linkers (such as '.cfi*'), no-ops ('nop'), brackets and commas are removed.

## A.2    Dataset Statistics

We present the detailed statistics of the datasets in Table 1 used in Evaluation Section.

**Table 1:** Statistics of the datasets benchmarks used in Evaluation Section.

| Length (tokens) | Karel$_{S/L}$ | Math$_{S/L}$ | NE$_{S/L}$ | (Math+NE)$_{S/L}$ | Pytorch | Hacker's light |
|---|---|---|---|---|---|---|
| Average output | 39/76 | 42/89 | 57/111 | 72/142 | 36 | 25 |
| Average input | 126/247 | 219/423 | 323/485 | 334/637 | 190 | 104 |
| Minimal variable number | 0 | 3/10 | 3/9 | 3/9 | 3 | 2 |
| Maximal variable number | 0 | 8/15 | 8/15 | 8/15 | 10 | 16 |
| branch program | ✓ | ✓ | ✓ | ✓ | ✗ | ✗ |

## A.3    Random Control Flow Graphs Generation

We evaluate various CFGs in our synthetic benchmarks. More specifically, multiple basic random CFGs (shown in Figure 1) are generated and connected to form the final CFG for the input program.

**Figure 1:** Examples of the possible control flow graph of dataset generation for Coda decompilation.

# B   Details of $FSM\_Error\_Correct$ Function

Algorithm 1 details the $FSM\_Error\_Correct$ function in Algorithm **??**. As mentioned in the Method Section, the Error Predictor (EP) outputs three error types, namely, (1) mispredicted tokens, (2) missing lines, and (3) redundant lines. For mispredicted error type, the Error Correction machine (EC machine) replaces the node token with the prediction output of the ensembled EP. As a special case of the mispredicted error where the terminal node is guided to become a non-terminal node, the EC machine adds random children to corresponding non-terminal node. As for missing errors, the EC machine adds extra a non-terminal node to the root node in the corresponding position, indicating the addition of a random newline to the output program. For the redundant error type, Coda removes the particular node from its parent. The structural difference between the mispredicted, missing and redundant errors is that the misprediction error occurs inside a program statement whereas the missing/redundant error indicates a lost/additional statement in the program. Coda's EC machine first transforms the missing/redundant error into the mispredicted error and then proceeds to iteratively fix the error insides the line.

---

**Algorithm 1** Correct Error FSM algorithm.

---

**INPUT: Decompiled program** $P^{'}$**; error type** $err$ **; Node id** $node$ **;** $\perp$ **terminal tokens; three types of error, viz, mispredicted** $pred$**, missing line** $ms$**, redundant line** $rdt$

**OUTPUT: Modified program** $P_t^{'}$**.**

1: **if** $err \in pred$ and $node \in \perp$ and $err \in \perp$ **then**
2:     $node.value \leftarrow err$
3: **else if** $err \in fp$ and $node \notin \perp$ and $err.value \notin \perp$ **then**
4:     $node.value \leftarrow err.value$
5: **else if** $err \in pred$ and $node \notin \perp$ and $err.value \in \perp$ **then**
6:     $node \leftarrow newNode()$
7:     $node.value \leftarrow err.value$
8: **else if** $err \in fp$ and $node \in \perp$ and $err \notin \perp$ **then**
9:     $node \leftarrow Add\_Random\_Children(node, \perp)$
    //assign random terminal children for the node
10:     $node.value \leftarrow err.value$
11: **else if** $err \in rdt$ **then**
12:     $node.parent.remove(node)$        //remove the node from its parent
13: **else if** $err \in ms$ **then**
14:     $node' \leftarrow newNode(not \perp)$
15:     $node' \leftarrow Add\_Random\_Children(node', \perp)$
16:     $node.parent.Add\_Children\_Behind(node', node)$ //add random non-terminal node to the root node behind the current node
17: **endif**
18: **Return:** $P_t^{'} \leftarrow P^{'}$

---

## C   Hyper-parameters of Neural Network Models

We present the hyper-parameters used by different neural networks in Table 2. These hyper-parameters are selected to achieve the best accuracy using cross-validation with *grid search*.

**Table 2:** Hyper-parameters chosen for each neural network model

|  | Seq2Seq | Seq2AST | Inst2AST | Inst2AST | EP |
|---|---|---|---|---|---|
| Batch Size | 50 | 50 | 50 | 100 | 10 |
| Number of RNN layer | 2 | 1 | 1 | 1 | 1 |
| Encoder RNN cell | LSTM | LSTM | N-ary LSTM | N-ary LSTM | - |
| Decoder RNN cell | LSTM | Tree LSTM | LSTM | Tree LSTM | GRU |
| Learning rate | Decay the learning rate by a factor of 0.9× when the validation loss does not decrease for 200 mini-batches | | | | |
| Hidden state size | 128 | | | | |
| Embedding size | 128 | | | | |
| Dropout Rate | 0.5 | | | | |
| Gradient clip threshold | 1.0 | | | | |
| Weight Initialization | Uniform Random from [-0.1,0.1] | | | | |

## D   Coda Decompilation Evaluation on x86-64 ISA

We also evaluate Coda's performance on x86-64 ISA and summarize the results in Table 3. We compile the code using `gcc -O0` configuration. Coda's decompilation token accuracy on x86-64 achieves is on average 6% lower than the one on MIPS architecture.

**Table 3:** Token accuracy across benchmarks on x86 assembly input.

| Benchmarks | Seq2Seq+Attn | Seq2AST+Attn | Inst2AST+Attn |
|---|---|---|---|
| Karel$_S$ | 96.73 | 99.50 | **99.61** |
| Math$_S$ | 90.16 | 96.19 | **96.50** |
| NE$_S$ | 85.73 | 88.76 | **89.33** |
| (Math+NE)$_S$ | 77.51 | 82.15 | **87.84** |
| Karel$_L$ | 95.20 | 96.17 | **96.41** |
| Math$_L$ | 86.64 | 91.55 | **92.60** |
| NE$_L$ | 78.56 | 80.63 | **83.19** |
| (Math+NE)$_L$ | 73.64 | 77.67 | **81.12** |

# E Examples Benchmarks Task

We present the examples in Figure 1 and 2. For simplicity, we list only the code snippet example for each benchmarks. The training dataset has different program length and variable numbers. For `Math+NE`, the dataset is build by replacing the variables in NE with functions.

**Figure 1:** Benchmark examples for (i) Pytorch C++ API (ii) Hacker's Delight

```cpp
// (i) NN construction for MNIST
struct nn::Module {
    Net()
      : conv1(Conv2dOptions(v1,v2,v3)),
        conv2(Conv2dOptions(v2,v4,v3)),
        fc1(v5, v6),
        fc2(v6, v7)
    }
    Tensor forward(Tensor x) {
    x = conv1->forward(x);
    x = torch::max_pool2d(x,2);
    x = torch::relu(x);
    x = conv2->forward(x);
    x = torch::max_pool2d(x,2);
    x = x.view({-1, v5});
    x = torch::relu(x, 2));
    x = fc1->forward(x);
    x = torch::relu(x);
    x = torch::dropout(x);
    x = fc2->forward(x);
    x = torch::log_softmax(x,1)
    return x;
  }
};

// (ii) Hacker's Delight example:
int32_t p25(int32_t x, int32_t y,\
int32_t base, int shift) {
  uint32_t o1 = x & base;
  int32_t o2 = x >> shift;
  uint32_t o3 = y & base;
  int32_t o4 = y >> shift;
  uint32_t o5 = o1 * o3;
  int32_t o6 = o2 * o3;
  int32_t o7 = o1 * o4;
  int32_t o8 = o2 * o4;
  int32_t o9 = o5 >> shift;
  int32_t o10 = o6 + o9;
  int32_t o11 = o10 & base;
  int32_t o12 = o10 >> shift;
  int32_t o13 = o7 + o11;
  int32_t o14 = o13 >> shift;
  int32_t o15 = o14 + o12;
  return o15 + o8;
}
```

**Figure 2:** Benchmark examples for (i) `Karel` (ii) `Math` (iii) NE. (iv) `Math+NE`.

```cpp
// (i) Karel
int main(){
    TurnOn();
    TurnOff();
    while(leftIsClear){
        PutBeeper();
        TurnLeft();
        if(notFacingNorth){
            continue;
        }
        PickBeeper();
        Move();
    }
    PickBeeper();
}
// (ii) Math
int func(int a, double b, int c, \
double d, double e){
    b=log(a);
    while(islessequal(d,a)){
        e=isgreaterequal(c,b);
        a=cos(e);
    }
    d=atan2(c,i);
    b=atan2(d,e);
    a=fmin(b,c);
}
// (iii) Normal Expressions
int func(int a, int b, double c) {
    a=a-b;
    c=b+c>>a;
    if((c>a)||(b<=e)){
        c=(c+a)/d;
        b=d*c-b+e;
        b=c<<a;
    }
}
// (iv) Math+Normal Expressions
int func(double a, int b, \
double c, double d, double e){
    b=log(a)-atan(d);
    while(isgreater(d,a)||isless(e,d)){
        e=isgreaterequal(c,b);
        a=cos(e);
    }
    d=atan2(c,i);
    b=(atan2(d,e)-fmax(c,a))/ceil(c);
    a=fmin(b,c)*asin(d)/pow(d,f);
}
```

# F  Examples of Traditional Decompiler Results

We present the decompiled results of traditional decompiler and Coda in Figure 3 and 4. Note that RetDec is the most recent published decompiler with more than 500K lines of code, which is 100× larger than Coda. Their toolkit size is ~5GB which is around 500× larger than the size of Coda's neural networks(~10MB).

**Figure 3:** Source Code and corresponding decompiled results from (i) Coda and (ii) RetDec. This example shows that the state-of-the-art decompiler fails to preserve the functionality and semantics. g1 to g7 are global variable that is used to pass parameters.

```
// source code
int code(double a, double b,\
double c){
    b=log(c);
    while(b<a){
        a=cos(b)*c;
    }
    return b;
}

// (i) decompiled code from Coda
int func(double v1, double v2, \
double v3){
    v2=log(v3);
    while(v2<v3){
        v1=cos(v2)*v3;
    }
    return v2;
}
// (ii) decompiled code from RetDec
int32_t code(void) {
    log((float64_t)(int64_t)g1);
    float64_t v1 = g6;
    if (g7 < v1) {
        float64_t v2 = v1 * v1;
        float64_t v3 = v1 * v2;
        float64_t v4 = v2 + v3;
        while (v3 < v4) {
            v1 = v4;
            v2 = v1 * v1;
            v3 = v1 * v2;
            v4 = v2 + v3;
        }
        g6 = v4;
    }
    __asm_cfc1(g8);
    int32_t v5 = __asm_cfc1(g8);
    __asm_ctc1(v5 & -4 | 1, g8);
    __asm_ctc1(v5, g8);
    return (float32_t)g6;
}
```

**Figure 4:** Source Code and corresponding decompiled results from (i) Coda and (ii) RetDec. The result shows that RetDec succeeds in preserving the functionality of the code in this case while the recovered high-level code is still difficult for human understanding.

```
// source code
int main(int argc, char *argv[]){
    int a = atoi(argv[1]);;
    int b = atoi(argv[2]);;
    int c = atoi(argv[3]);;
    a = b * c - 1 ;
    if( a > 1 ){
        a = b + c;
        c = a * c - b;
    }
    return c;
}
// (i) decompiled code from Coda
int main(int argc, char *argv[]){
    int v1 = atoi(argv[1]);;
    int v2 = atoi(argv[2]);;
    int v3 = atoi(argv[3]);;
    v1 = v2 * b3 - 1 ;
    if( v1 > 1 ){
        v1 = v2 + v3;
        v3 = v1 * v3 - v2;
    }
    return c;
}
// (ii) decompiled code from RetDec
int main(int argc, char ** argv) {
    int32_t v1 = (int32_t)argv;
    atoi((char *)*(int32_t *)(v1 + 4));
    int32_t v2 = *(int32_t *)(v1 + 8);
    int32_t v3 = *(int32_t *)(v1 +12);
    int32_t result;
    if (v3 * v2 >= 3) {
        result = (v3 + v2) * v3 - v2;
    } else {
        result = v3 * v2 < 3;
    }
    return result;
}
```