[Reviews · NeurIPS 2019]

Reviewer 1



The setting in which the paper operates is given a compiler T, compiler options O and output machine code M of a function to find source code S that compiles to M using T and O. This is not a particularly practically compelling case, because it usually breaks with compiler optimizations, but still it is a good start. The main idea of the paper, however, is an elegant process of first generating a "sketch" program which may not necessarily be correct on the first shot, but is then iterated until it is correct. Because the compiler is present as an oracle, it is always possible to check if the correctness requirements are satisfied. This process is a good idea, because the translation process from machine to source code needs to be always right, but a one-shot neural generation procedure is unlikely to capture the entire semantics properly and will inevitably output some errors. I expect the of numerical (or string) constants would be difficult to capture (Q1).

Reviewer 2



The paper is thought-provoking -- especially in the way it identifies errors and then uses brute-force search to look for solutions that minimize distance in the compilation space. The method is not pretty, as the brute-force search through the high level language space could certainly be improved. The paper throughout is frequently unclear and confusing to read. How do the two LSTMs for the left and right subtree children work? How exactly is the error prediction module trained? The paper makes a distinction between functionally preserving the code, and also semantically preserving the code. In this context, program language semantics has a distinct meaning -- another word should be used and this concept clarified. There are too many acronyms that make the paper difficult to read. e.g. LD for Levenshetin distance is not a common or well known acronym. It would be useful for the authors to rescheck their answers on the reproducibility checklist.

Reviewer 3



I have questions about the motivation of this work. Good, totally accurate decompilers exist already, and have for a very long time. The authors argue that the reason we need such a neural decompiler is that the development time is so much less… it takes a lot of time and effort to develop a classical decompiler. That may be true, but the decompiler described in this paper has the significant disadvantage of not being “correct” in the sense that its programs are not guaranteed to compile: they may have type errors, variable use without declaration errors, or semantic errors that are specific to the PL being decompiled into. This is a significant drawback compared to a classical decompiler, and the authors need to address this issue in the paper by answering the question: what are the applications where it is helpful to decompile into code that (possible) has all sorts of semantic errors, and why? Along those same lines, the paper needs to report the fraction of decompiled codes that have compilation errors. In general, the evaluation could be improved significantly. Really, the main results given are in Table 1, which gives the “token accuracy” of the various methods for recovering the original code. First, it is not clear to me what this actually means. Does this mean that if I lex the two programs (original and recovered) that this is the fraction of tokens that match? Why is this a meaningful metric? If I have even one token wrong, couldn’t this fundamentally change the semantics of the program that I have recovered? It seems that a good decompiler should have two main requirements. First, the semantics should match the original code. Second, the decompiled code should somehow look similar (the original program should be structured the same way as the original program, variable names should be meaningful and hopefully match the original, etc.). “Token accuracy” at least somewhat addresses the second, but the semantic correctness is not addressed at all in the paper. The authors do define the notation of “program accuracy” (100% token accuracy) but as far as I can tell, that is not given in the paper. And, as I mentioned above, metrics such as the percentage of output codes that actually compile are not given. It would be really helpful to see how well things work when the programs get longer. The programs that are being decompiled are all very short… “long” programs have length 30 (is this 30 lines of assembly?). That does not seem to be very long. It’s a little unclear to me whether the programs being decompiled are all simple, straight line codes, or whether they include more complicated structures (such as static global variables, classes, function pointers, function calls, function bodies, etc.). Obviously, if the paper is only dealing with programs that do not have these structures, the methods being developed are a lot less useful. I found it strange that in a paper about decompilation, no example codes are given! Why don’t the numbers in Tables 1 and 2 match? That is, I would expect some of the numbers to be the same across the two tables, because you are testing the same thing. But this does not appear to be the case. Why?

[Author Response · NeurIPS 2019]

We thank all the reviewers (R4,R5,R6) for the valuable feedback.

**Difference between semantics and functionality (R6 & R5).** Functionality (or correctness) of the code describes
whether the behavior of the recovered code matches the low-level code. Semantics (or readability) is defined as the
similarity between the recovered code and original high-level (HL) code. Our experiments (Appendix F, line 324-331)
show that existing decompilers hardly satisfy neither functionality nor semantics. Their recovered programs are hard
to interpret and dissimilar to the source programs. Assembly code may be incorrectly translated into the assembly
operations in HL code. For example, mov instructions are likely to be translated into variable copy which is apparently
redundant. Note that the semantics of the generated HL program is insensitive to recovered token while the functionality
can be heavily affected (R6).

**Metric selection/Code examples/Table explanation (R6).** Token accuracy is similar to the BLEU score for NLP
evaluation. It shows how close the decompiled program and the original HL program are. Program accuracy measures
the percentage of recovered programs that preserve both functionality and semantics. These metrics are also used
in other program translation or synthesize works ([14, 36]). We provide code examples in supplementary materials
(Appendix E, F). Table 1 shows the **token accuracy** after Stage 1. The final **program accuracy** after Stage 2 is reported
in Table 2. As such, the results in these two tables shall not match.

**Key Motivations (R6).** Our key motivations: i) Our *end-to-end* design makes the decompilation task more efficient and
extensible. With the growing amount of new programming languages (PLs), new PL features (e.g. software obfuscation)
and various hardware (TPU/GPU/FPGA/Accelerators), current decompilers are very limited in their usage and incur
high engineering overhead. ii) Coda maintains both the functionality and semantics of the original HL program.

**Solutions to syntax errors (R6).** Coda trains the auto-encoder to let the neural network learn the grammar of the HL
language (line 41) without linguistics knowledge. Our result (Table 1) does not show the significant disadvantage of
containing a lot of syntax errors. We encounter only a few variable usage errors that lead to decompilation failure
because Coda automatically learns the number of variables and their types. For AST decoding method, the generated
AST is guaranteed to be compilable. A small portion of syntax errors exists in the sequence decoding baseline and
we use a script to check and fix these syntax bugs. The numbers in Table 1 are measured before the script checking.
Also, the error correction stage guarantees that the sketch code from stage 1 is fault-tolerant. Other errors that do not
influence the compilation can be corrected in the second stage.

**Recover complicated structures (R6).** We use similar benchmarks as the previous decompiler works [Phoniex
USENIX'13], including function calls, normal expressions, nested control graphs, variables with different types
and data dependencies. Note that existing decompilers also fail to recover complicated data structures/classes. The
type and structure identification is an individual research direction which has been widely studied in previous works
[3][REWARD NDSS'10]. Coda is sufficient to resolve real-world applications such as Pytorch API or Hacker's delight
applications. Combining Coda with these works can recover more complicate programs.

**Unclear methods (R5): 1) AST tree decoder.** The states ($h$,$c$) from a given AST node will feed into the left/right
LSTM to generate the left/right child, as shown in Eq. (2) and (3) (line 201-210). These two expanded nodes will
become the new parent nodes to generate its children using the left/right LSTM. The obtained binary AST tree (left-child
right-sibling representation) will be transferred back to its equivalent AST tree. **2) Error Predictor Training.** The
training target is the error type of the given AST node that we manually injected. If the error type for the given node is a
mispredicted error, the EP also outputs the correct substitution token (line 232). The training loss between the EP's
outputs and the targets is minimized. The training data is fixed during the EP's training.

**Experiment with numerical representation (R4):** The numerical representation is an existing issue in NLP appli-
cations and solutions have been proposed in other works. In our case, most of the numeric is the offset addresses
that appear frequently. We treat them the same way as other tokens. The numeric can also be represented as a
real-value scalar. Our experiments show for small memory footprint programs (number of variables = 10), the numerical
representation in different encoding format does not lead to significantly different performance (Scalar encoding: +0.9%
program accuracy on average). With an increasing number of variables and memory usage (variables = 20), scalar
format shows more scalability (+2.7% program accuracy).

**Experiment with longer codes (R6):** With average code length ($L$) of 45/60, the token accuracy drops by (seq2seq:
-5.4%/-13.5%,inst2ast: -3.1%/-8.4%) on average compared to $L = 30$ across benchmarks. For longer codes, instruction
encoding shows better performance compared to seq2seq model. The challenges to decompile long programs are: i)
Unlike natural language with period as the end of sentence, there is no clear boundary to divide assembly code. The
length of the input tokenized assembly grows to a very large value (Appendix A.2). One possible solution to this
problem is to divide the code using function entry point. ii) the GPU memory is not enough to train the network with a
large batch size for tasks with extremely long encoding sequences.

[Meta-Review · NeurIPS 2019]

There was a significant disagreement between the reviewers on this paper. All of the reviewers thought that the search phase to repair the original decompilation was an intriguing contribution. The main difference of opinion was on the difficulty of the examples presented in this paper, and of C decompilation in general. To this end, I note that the main competitor that is compared to (retdec) is based on a recent PhD thesis (J. Křoustek: Retargetable Analysis of Machine Code. Ph.D. Thesis, FIT BUT, Brno, CZ, 2015, pp. 190 --- I would suggest to the authors of this submission that it would be kind to cite the thesis in addition to the URL), and the supplemental material shows a pretty convincing case where this method breaks down. I also note that several of the papers that the authors cite show results of 80-98% (so less than perfect) on the subtask of identifying function boundaries in stripped C executable --- so just figuring out **where** to decompile, and not actually producing code. This also seems like clear evidence that for C code (unlike Java bytecode, which is much easier) decompilation is still a difficult open issue.